# HOW PRIVATE IS DIFFUSION-BASED SAMPLING?

## ABSTRACT

Diffusion models have emerged as the foundation of modern generative systems, yet their high memorization capacity raises privacy concerns. While differentially private (DP) training provides formal guarantees, it remains impractical for large-scale diffusion models. In this work, we take a different route by analyzing privacy leakage during the sampling process. We introduce an empirical denoiser that enables tractable computation of per-step sensitivities, allowing each denoising step to be interpreted as a Gaussian mechanism. Building on this perspective, we apply Gaussian Differential Privacy (GDP) to derive tight privacy bounds. Furthermore, we identify critical windows in the denoising trajectory—time steps where salient semantic features emerge—and quantify how privacy loss depends on stopping relative to these windows. Our study provides the first systematic characterization of privacy guarantees in diffusion sampling, offering a principled foundation for designing privacy-preserving generative pipelines beyond DP training.

## 1 INTRODUCTION

Diffusion models (Sohl-dickstein et al., 2015; Ho et al., 2020a) have rapidly become the foundation of modern generative systems, powering applications in Stable Diffusion (Rombach et al., 2022a) and proprietary models like Flux (Labs et al., 2025). At their core, these models simulate a *denoising diffusion process*, where a forward diffusion process gradually corrupts an observed sample and a reverse denoising process inverts this corruption. The forward and backward processes are typically modelled as a Gaussian Markov chain for tractability, where the forward process is parameter-free, simply adding noise to a sample following a predefined noise schedule. On the other hand, neural networks, often referred to as *neural denoisers*, are typically used to learn the reverse denoising process for sampling in diffusion models, enabling high-quality generations.

While highly effective, however, diffusion models are also known for their high memorization capacity. Generated samples are often exact replicas of the training data, raising critical privacy concerns (Wu et al., 2023; Carlini et al., 2023; Tang et al., 2023; Hu & Pang, 2023; Duan et al., 2023a; Matsumoto et al., 2023). To mitigate this issue, a range of strategies have been explored. Post-hoc methods focus on detecting or filtering memorized outputs after training (Wen et al., 2024), whereas knowledge distillation and architectural modifications aim to reduce the model's inherent tendency to memorize by altering how knowledge is transferred or represented (Gu et al., 2023). These approaches can meaningfully reduce privacy risks, but they often lack rigorous guarantees.

A more principled solution is to employ differentially private (DP) (Abadi et al., 2016) for diffusion training (Dockhorn et al., 2023; Ghalebikesabi et al., 2023; Liu et al., 2024). Differential privacy provides a formal guarantee that the influence of any single training example on the final model is limited. Intuitively, this means that whether or not a particular user's data is included in the training set, the model's behavior should remain (almost) the same. Thanks to DP's post-processing invariance property, any samples generated from a diffusion model guaranteed by DP inherit formal privacy guarantees to a quantifiable extent.

Beyond its mathematical rigor, DP has become the de facto standard for formal privacy protection in practice. For example, the U.S. Census Bureau adopted DP in the 2020 Census to guarantee that published statistics do not reveal whether any individual's data was included (Abowd, 2018). Likewise, data protection frameworks such as General Data Protection Regulation (GDPR) from EU and California Consumer Privacy Act (CCPA) increasingly demand verifiable guarantees that individual contributions cannot be inferred. In this sense, DP is not merely a theoretical construct

but a policy-relevant framework that directly addresses real-world privacy concerns. Despite these appealing guarantees, however, DP training algorithms remain largely impractical for large-scale models, since protecting every data points requires injecting significant noise at each parameter update, which severely degrades generation quality.

In this paper, we take a different route to the privacy issues of diffusion models. Rather than proposing a new DP training algorithm, we aim to provide a systematic study of **privacy loss incurred during the sampling process**. Given that current DP-based training for diffusion models has not achieved satisfactory performance, we shift the focus from the model itself to the generated samples. This change in perspective is both natural and practical: end users and regulators ultimately interact with the outputs, and in many proprietary systems the model weights are not released. Thus, while model training may not always need to be differentially private, the samples it produces should be. The challenge, however, lies in the fact that performing privacy analysis on neural denoisers is extremely difficult. Specifically, computing the sensitivity of a trained neural denoiser—defined as the maximum change in its output when a single training example is modified—is intractable.

To resolve this issue, we introduce an *empirical denoiser* that allows us to compute the *sensitivity* of each denoising step. Each sampling step adds stochastic Gaussian noise, which naturally aligns with the well-studied Gaussian mechanism in differential privacy. This observation suggests that the entire sampling process can be analyzed as a composition of Gaussian mechanisms, a perspective that enables us to apply existing DP theory in a principled way.

A central challenge, however, is how to track the cumulative privacy loss across many denoising steps. The classical $(\epsilon, \delta)$-DP framework provides only loose bounds when composing a large number of Gaussian mechanisms, leading to overly pessimistic privacy estimates. To address this, we adopt the framework of *Gaussian Differential Privacy (GDP)*, which replaces the pair $(\epsilon, \delta)$ with a single parameter $\mu$ that more tightly characterizes the privacy profile of Gaussian mechanisms. Intuitively, GDP plays a role for privacy accounting similar to the Gaussian distribution in probability theory: just as the central limit theorem shows that sums of independent random variables converge to a Gaussian, the composition of many private mechanisms converges to GDP (Dong et al., 2022). This makes GDP not only yield sharper composition guarantees but also a natural fit for analyzing diffusion sampling, where Gaussian perturbations are intrinsic at every step.

However, the key factor determining the validity of this study is how closely the empirical denoiser aligns with the neural (or ground-truth) denoiser. The reliability of our results hinges on this approximation, which is most accurate during the middle to late stages of the diffusion process. Accordingly, we evaluate privacy within this window, which notably coincides with the *critical phases* where salient image features begin to emerge. This carries an important implication: our analysis quantifies privacy leakage precisely in the interval where semantic content arises. A natural question then concerns the remaining denoising steps that refine the image to high fidelity. Since our approximation becomes less reliable in this regime, we conjecture that these later steps could instead be performed with a diffusion model trained on public (and non-private) data, thereby preserving privacy once the semantic structure has already been established.

Our contributions are summarized below:

1. We formulate diffusion sampling as a composition of Gaussian mechanisms and show how to compute per-step sensitivities.

2. We apply GDP analysis to derive tight privacy bounds for diffusion sampling under varying noise schedules, for both full-batch and mini-batch settings.

3. We extend this analysis to identify *critical windows*—time steps where salient image features emerge—and quantify the privacy level depending on where we stop in relation to the critical window of the denoising process.

## 2 BACKGROUND

### 2.1 DENOISING DIFFUSION MODELS

Diffusion models are built upon the forward and backward processes (Ho et al., 2020b; Song et al., 2020). The forward perturbation describes how the noise is gradually corrupting data sample $\mathbf{x}_0$,

defined as $\mathbf{x}_t = \alpha_t \mathbf{x}_0 + \sigma_t \boldsymbol{\epsilon}$, where $\boldsymbol{\epsilon}$ is a standard Gaussian variable, $\alpha_t$ and $\sigma_t$ are prescribed time-dependent mean and variance coefficients, respectively. For simplicity, we describe the diffusion process from EDM (Karras et al., 2022), but our framework is adoptable to any diffusion processes such as DDPM (Ho et al., 2020a) or Flow Matching (FM) (Lipman et al., 2022). In EDM, the forward diffusion process is simply defined by $\mathbf{x}_t = \mathbf{x}_0 + t\boldsymbol{\epsilon}$, i.e., $\alpha_t \equiv 1$ and $\sigma_t = t$, and the reverse denoising process is computed by

$$\mathbf{x}_{t-\Delta t} = \left(1 - 2\frac{\Delta t}{t}\right)\mathbf{x}_t + 2\frac{\Delta t}{t}\mathbb{E}[\mathbf{x}|\mathbf{x}_t] + \sqrt{2t\Delta t}\boldsymbol{\epsilon}. \tag{1}$$

The term $\mathbb{E}[\mathbf{x}|\mathbf{x}_t]$, called a Bayes optimal denoiser, is the conditional expectation of the clean data given the noisy observation $\mathbf{x}_t$. In practice, a neural denoiser $D(\mathbf{x}_t, t; \boldsymbol{\theta}(\mathcal{D}))$ approximates this quantity, where $\boldsymbol{\theta}(\mathcal{D})$ are parameters learned from dataset $\mathcal{D}$. By rolling out eq. 1 up to time zero, diffusion model gradually synthesizes a sample.

## 2.2 Differential Privacy and Gaussian Mechanism

Differential Privacy (DP) provides a formal framework to limit the influence of any single record on the output of a randomized algorithm. Intuitively, whether or not an individual's data is present, the distribution of outputs should remain (almost) unchanged. This view naturally extends to diffusion sampling, which is itself a randomized mechanism.

A fundamental building block is the *Gaussian mechanism*: given a function $h : \mathcal{D} \mapsto \mathbb{R}^p$, the mechanism releases

$$\mathcal{M}_{\text{Gauss}}(\mathcal{D}; h) = h(\mathcal{D}) + \boldsymbol{n}, \qquad \boldsymbol{n} \sim \mathcal{N}(0, \sigma^2 \Delta_h^2 \mathbf{I}), \tag{2}$$

where $\Delta_h$ is the global sensitivity—the maximum change in $h$ between neighboring datasets. By calibrating the noise to $\Delta_h$, the Gaussian mechanism guarantees DP (see Appendix B.1 for formal definitions).

For analyzing multi-step procedures such as diffusion sampling, we adopt *Gaussian Differential Privacy (GDP)* (Dong et al., 2022). GDP tightly characterizes the privacy loss of Gaussian mechanisms and, crucially, admits an exact composition rule:

$$\mu = \sqrt{\sum_i \mu_i^2}, \tag{3}$$

where $\mu_i$ is the GDP parameter of the $i$-th step. This makes GDP especially well suited for diffusion processes, which inherently inject Gaussian noise at every denoising step.

Two further facts are important for our analysis. First, privacy amplification by subsampling can further reduce the effective $\mu$. Second, GDP parameters can be translated into $(\epsilon, \delta)$ guarantees when needed for regulatory reporting; we provide details in Appendix B.2. Together, these tools allow us to connect the denoising process directly to established DP accounting methods.

# 3 Privacy Analysis of Denoising Process with Empirical Denoiser

## 3.1 Single-Step Denoising is a GDP

As explained in the background section, the denoiser is defined as the conditional expectation

$$\mathbb{E}[\mathbf{x}|\mathbf{x}_t] = \frac{1}{p_t(\mathbf{x}_t)} \int \mathbf{x}_0 \, p(\mathbf{x}_t|\mathbf{x}_0)p(\mathbf{x}_0) \, \mathrm{d}\mathbf{x}_0.$$

In practice, this quantity is not computed explicitly but is approximated by a neural network

$$D(\mathbf{x}_t, t; \boldsymbol{\theta}(\mathcal{D})) \approx \mathbb{E}[\mathbf{x}|\mathbf{x}_t],$$

where the parameters $\boldsymbol{\theta}(\mathcal{D})$ are learned from dataset $\mathcal{D}$. However, analyzing privacy directly through the neural denoiser would require bounding its global sensitivity, $\Delta = \sup_{\mathbf{x}_t} \|D(\mathbf{x}_t, t; \boldsymbol{\theta}(\mathcal{D})) - D(\mathbf{x}_t, t; \boldsymbol{\theta}(\mathcal{D}'))\|$, for neighboring datasets $\mathcal{D}, \mathcal{D}'$. Since $D(\cdot)$ is a deep network with millions or

billions of parameters that change in a highly non-linear fashion with respect to the training data, computing or even meaningfully bounding $\Delta$ is intractable.

To circumvent this issue, we introduce an *empirical denoiser* $\hat{\mathbb{E}}[\mathbf{x}|\mathbf{x}_t; \mathcal{D}]$, replacing the conditional expectation with a dataset-based approximation:

$$\mathbb{E}[\mathbf{x}|\mathbf{x}_t] \approx \hat{\mathbb{E}}[\mathbf{x}|\mathbf{x}_t; \mathcal{D}] = \frac{1}{p_t(\mathbf{x}_t)|\mathcal{D}|} \sum_{\mathbf{x}_0 \sim \mathcal{D}} \mathbf{x}_0 \, p(\mathbf{x}_t|\mathbf{x}_0).$$

Plugging this empirical denoiser into eq. 1, the denoising process becomes

$$\mathbf{x}_{t-\Delta t} = \left(1 - 2\frac{\Delta t}{t}\right) \mathbf{x}_t + 2\frac{\Delta t}{t} \frac{1}{p_t(\mathbf{x}_t)} \left(\frac{1}{|\mathcal{D}|} \sum_{\mathbf{x}_0 \sim \mathcal{D}} \mathbf{x}_0 p(\mathbf{x}_t|\mathbf{x}_0) + p_t(\mathbf{x}_t)\sqrt{\frac{t^3}{2\Delta t}}\boldsymbol{\epsilon}\right).$$

Now consider the expression inside the parentheses:

$$\frac{1}{|\mathcal{D}|} \sum_{\mathbf{x}_0 \sim \mathcal{D}} \mathbf{x}_0 p(\mathbf{x}_t|\mathbf{x}_0) + p_t(\mathbf{x}_t)\sqrt{\frac{t^3}{2\Delta t}}\boldsymbol{\epsilon} \tag{4}$$

This takes the form of a Gaussian mechanism, where we apply a norm clipping approach (Abadi et al., 2016) to bound sensitivity. Specifically, let $\mathbf{v}(\mathbf{x}_0; \mathbf{x}_t) := \mathbf{x}_0 p(\mathbf{x}_t|\mathbf{x}_0)$, and define

$$\bar{\mathbf{v}}(\mathbf{x}_0; \mathbf{x}_t) := \frac{\mathbf{v}(\mathbf{x}_0; \mathbf{x}_t)}{\max\left(1, \frac{\|\mathbf{v}(\mathbf{x}_0; \mathbf{x}_t)\|}{C}\right)},$$

where $C$ is a threshold for the norm as explained next. When $\|\mathbf{v}(\mathbf{x}_0; \mathbf{x}_t)\| < C$, we have $\bar{\mathbf{v}} = \mathbf{v}$, and when it exceeds $C$, the vector is scaled to have norm $C$. Substituting the clipped version $\bar{\mathbf{v}}$ into eq. 4, we obtain

$$\mathcal{M}_{\text{Gauss}}(\mathcal{D}; \mathbf{x}_t) = \frac{1}{|\mathcal{D}|} \sum_{\mathbf{x}_0 \sim \mathcal{D}} \bar{\mathbf{v}}(\mathbf{x}_0; \mathbf{x}_t) + p_t(\mathbf{x}_t)\sqrt{\frac{t^3}{2\Delta t}} \, \boldsymbol{\epsilon}. \tag{5}$$

As the sensitivity of the summation over $\bar{\mathbf{v}}$ is bounded by $\frac{2C}{|\mathcal{D}|}$, according to eq. 2, eq. 5 corresponds to $\frac{2C}{p_t(\mathbf{x}_t)|\mathcal{D}|}\sqrt{\frac{2\Delta t}{t^3}}$-GDP. In conclusion, our denoising process can be viewed as $\mu_t$-GDP at each step, where $\mu_t := \frac{2C}{p_t(\mathbf{x}_t)|\mathcal{D}|}\sqrt{\frac{2\Delta t}{t^3}}$.

## 3.2 MULTI-STEP DENOISING IS ALSO A GDP

Building on this single-step characterization, we next extend the analysis to multi-step denoising. Suppose that we denoise the sample through discretized timesteps $t_i$ with $t_i < t_{i+1}$, $t_0 = \sigma_{min}$ and $t_N = \sigma_{max}$. The gap $\Delta t$ is now depending on the timestep index with $\Delta t_i = t_i - t_{i-1}$ and the GDP level for each timestep is $\mu_{t_i} = \frac{2C}{p_t(\mathbf{x}_t)|\mathcal{D}|}\sqrt{\frac{2\Delta t_i}{t_i^3}}$. The multi-step denoising is equivalent to applying GDP mechanisms iteratively. According to eq. 3, if we consider a mechanism $\mathcal{M}$ to be a denoising process from $t_m$ to $t_n$ (given $m > n$), then this mechanism is also a GDP with $\mu = \sqrt{\sum_{i=n}^{m} \mu_{t_i}^2}$.

Knowing the $\mu$ value at each step is not just a technicality—it provides an interpretable measure of privacy leakage. In the GDP framework, $\mu$ fully characterizes the trade-off curve of hypothesis testing between two neighboring datasets, describing exactly how well an adversary could distinguish whether an individual's data was present. Smaller $\mu$ values correspond to stronger privacy guarantees, meaning the adversary cannot reliably distinguish between the two cases; larger $\mu$ values correspond to weaker guarantees, as the adversary's distinguishing power increases.

Crucially, because $\mu$ values accumulate across timesteps, the total $\mu$ grows as sampling proceeds. This means that the early steps of denoising—when the sample is still close to pure noise—offer stronger privacy protection, while the later steps—when semantic structure becomes more pronounced—yield weaker guarantees. In this way, tracking how $\mu$ evolves over time allows us to pinpoint when privacy is strongest, when it degrades, and how the overall guarantee emerges across the entire sampling trajectory.

### 3.3 BRIDGING EMPIRICAL AND NEURAL DENOISERS

A central limitation of our analysis is that real-world diffusion sampling relies on neural denoisers, whereas our privacy characterization is derived using an empirical denoiser. However, directly analyzing neural denoisers is infeasible, since their sensitivity cannot be tractably computed. This discrepancy inevitably introduces some inaccuracy in estimating real-world privacy leakage. The precise impact of this gap on privacy accounting remains an open problem, and we view it as an important direction for future research.

That said, there are reasons to believe our analysis is conservative. To see this, recall the bias–variance decomposition:

$$\mathbb{E}_{\mathcal{D}}[\|\hat{f} - \mathbb{E}[\mathbf{x}|\mathbf{x}_t]\|_2^2] = \underbrace{\|\mathbb{E}_{\mathcal{D}}[\hat{f}] - \mathbb{E}[\mathbf{x}|\mathbf{x}_t]\|_2^2}_{\text{Bias}^2} + \underbrace{\mathbb{E}_{\mathcal{D}}[\|\hat{f} - \mathbb{E}_{\mathcal{D}}[\hat{f}]\|_2^2]}_{\text{Variance}}.$$

As a heuristic argument, consider two possible estimators $\hat{f}$: the empirical denoiser $\hat{\mathbb{E}}[\mathbf{x}|\mathbf{x}_t; \mathcal{D}]$ and the neural denoiser $D(\mathbf{x}_t, t; \boldsymbol{\theta}(\mathcal{D}))$. If we suppose that the mean squared error of this two estimators relative to the population denoiser $\mathbb{E}[\mathbf{x}|\mathbf{x}_t]$ is equal, then the bias–variance decomposition gives us an insight: since the empirical denoiser $\hat{\mathbb{E}}$ is an unbiased estimator of the population denoiser $\mathbb{E}$, its bias term is zero, and thus all of its MSE must come from variance. In contrast, the neural denoiser may incur some bias and can achieve lower variance.

This implies that the empirical denoiser has a larger variance, and therefore greater sensitivity, than the neural denoiser. In other words, our analysis is more likely to exaggerate than to downplay the amount of privacy leakage that would occur with real neural denoisers. This gives some reassurance that our privacy analysis is rather conservative. At the same time, this reasoning relies on the simplifying assumption that the empirical and neural denoisers achieve similar accuracy, which may not always hold in practice.

To provide empirical support, Figure 1 compares the denoising directions produced by empirical and neural denoisers. While early timesteps may still differ, Figure 1 shows that in later timesteps the denoising directions are nearly indistinguishable.

Theoretically, recent papers (Li & Chen, 2024; Biroli et al., 2024; Sclocchi et al., 2024) suggest that there are a few discrete phase transitions occurring during the sampling process, in which a generating image's class membership in an unconditional sampling is determined at a relatively early stage in the denoising process. Once determined, the image's membership remains the same until the end of the sampling process. These theoretical findings are supported by empirical papers such as (Georgiev et al., 2023).

These papers suggest that the samples by neural denoisers form the large distinguishing features at $t \in [T, t_c]$ for some $t_c \gg 0$, where our empirical denoiser and the neural denoiser behave similarly. Hence, it is sensible to study the privacy implications of these two denoisers in the *formative* stage where they are relatively similar to each other.

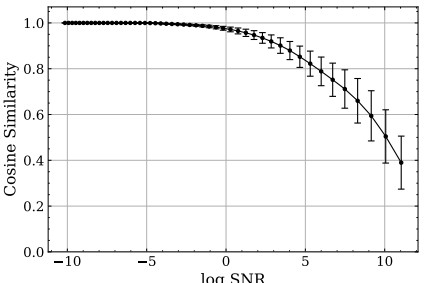

Figure 1: Cosine similarity between the denoising directions, i.e., $\text{Cos}(\mathbf{x}_t - \hat{\mathbb{E}}, \mathbf{x}_t - D)$, by empirical and neural denoisers. We report results along the log-SNR axis (Kingma et al., 2021), where the Signal-to-Noise Ratio is defined as $\text{SNR} = 1/4t^2$. SNR provides a unified way to represent the noise scale across different diffusion formulations (e.g., EDM, DDPM, and FM), allowing our analysis to be interpreted independently of a specific noise schedule.

We also numerically evaluate the privacy implications of these two denoisers in terms of their attack success rate on the membership inference attacks (MIAs) in Sec. 6.

### 3.4 PRACTICAL APPLICATIONS BY PRIVACY ISOLATION

As discussed in Section 3.2, the cumulative $\mu$ value increases as denoising proceeds, meaning that privacy guarantees become weaker as we denoise more steps. This observation suggests a practical strategy: one could only apply the privacy-sensitive denoising when semantic structure emerges (at

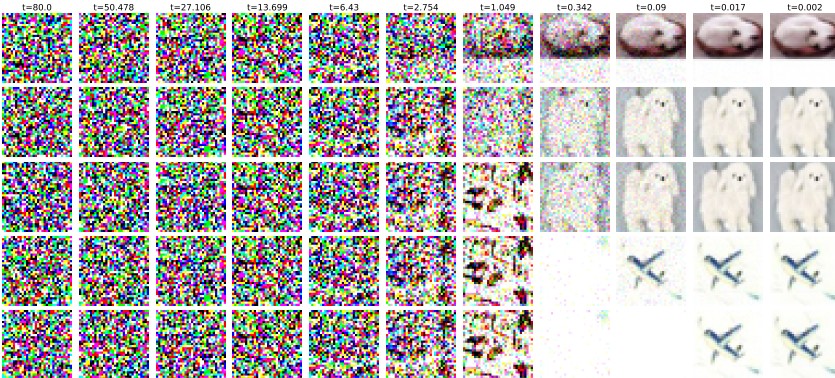

Figure 2: Example samples using our practical applications described in Section 3.4. A neural denoiser is applied first to $t = 6.43, 2.754, 1.049, 0.342, 0.09$. Then, we finish diffusion with our empirical denoiser

the medium time regime), and switch to a non-private diffusion model trained on public data to the remaining steps (early/late time regime). Because the early and late steps do not contribute to the overall privacy accounting, this hybrid approach may provide strong privacy guarantees without sacrificing final sample quality. Figure 2 illustrates generated samples under different $\epsilon$ values, supporting the feasibility of this approach.

## 4 PRIVACY ANALYSIS OF DENOISING PROCESS WITH SUBSAMPLED EMPIRICAL DENOISER

So far, our analysis in Section 3 treated the empirical denoiser in a full-batch setting, where every record in $\mathcal{D}$ contributes to each denoising step. An interesting question that may arise is how the privacy analysis changes if this empirical denoiser is evaluated on randomly subsampled datapoints? In the case of diffusion sampling, how much does subsampling amplify privacy when using our empirical denoiser?

Recall from Section 3 that a single denoising step under the empirical denoiser can be written as a Gaussian mechanism,

$$\mathcal{M}_{\text{Gauss}}(\mathcal{D}; x_t) = \frac{1}{|\mathcal{D}|} \sum_{x_0 \sim \mathcal{D}} \bar{v}(x_0; x_t) + p_t(x_t)\sqrt{\tfrac{t^3}{2\Delta t}}\,\epsilon.$$

When the dataset is replaced by a random subsample $\mathcal{S} \subseteq \mathcal{D}$ of expected size $q|\mathcal{D}|$, the mechanism becomes

$$\mathcal{M}_{\text{SubGauss}}(\mathcal{D}; x_t) = \frac{1}{|\mathcal{S}|} \sum_{x_0 \sim \mathcal{S}} \bar{v}(x_0; x_t) + p_t(x_t)\sqrt{\tfrac{t^3}{2\Delta t}}\,\epsilon.$$

The extra randomness from subsampling amplifies privacy: with probability $1 - q$, a record does not participate at all in the update, thereby reducing sensitivity.

Formally, this phenomenon is known as *privacy amplification by subsampling*. The $f$-DP framework (Dong et al., 2022) precisely characterizes the trade-off curve of the subsampled Gaussian mechanism. As we compose many steps, these curves converge to a Gaussian profile by a central limit effect. The trade-off function, however, is no longer a simple expression as was in full-batch setting, and it is necessary to compute $\mu$ numerically due to the varying amount of noise at each sampling step. See Appendix A for more details. We refer to this analysis as *GDP CLT* analysis, following Dong et al. (2022).

One might criticize our choice of using the CLT approach to view the composition of $f$-DP mechanisms as GDP asymptotically. In particular, under CLT, it is commonly assumed that no single mechanism makes a significant contribution to the composition in the limit. However, as we approach the end of the denoising process, the noise scale is significantly reduced, and thus the corresponding $f$-DP mechanism contributes more to the final $\mu$. This problem can be alleviated by using

the empirical denoiser in the time range where the noise scales are relatively similar to each other. For instance, if we use a neural denoiser trained with public data until some changing point $t^*$ (the neural denoiser does the majority of the denoising) and then switch to the empirical denoiser to sample a large number of samples (e.g., $10,000$ samples). Then, the noise scales at those timesteps are relatively similar to each other as shown in Fig. 3, avoiding any one term dominating others. With these relatively small and similar noise scales, drawing a large number of samples approaches the asymptotic regime in which the GDP CLT analysis was performed.

## 5 RELATED WORK

Differentially private training has been the most direct approach to reduce memorization in diffusion models. Building on DP-SGD (Abadi et al., 2016), recent studies have attempted to train diffusion models with differential privacy guarantees (Dockhorn et al., 2023). While these methods provide formal protection at the training stage, they perform poorly compared to non-private counterparts, as the noise required for every parameter update severely degrades sample quality.

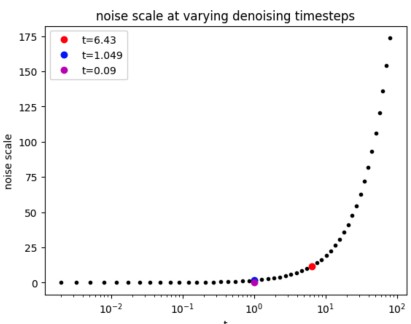

Figure 3: **Noise scale at varying denoising timesteps.** The noise scale drastically increases $t \to T$. However, as $t \to 0$, the noise scale remains relatively constant and small. For example, the noise scale at $t = 6.43$ is 11.77, at $t = 1.049$, it is 1.70, and at $t = 0.09$, it is 0.12.

Beyond training, the DP literature has also investigated privacy guarantees of sampling algorithms, though this direction has not yet been systematically explored for diffusion models. Most prior work arises in Bayesian posterior inference, where the goal is to sample parameter values from the posterior distribution rather than data samples from the data distribution. For example, Wang et al. (2015) showed that Stochastic Gradient Langevin Dynamics (SGLD) (Welling & Teh, 2011) is differentially private with sufficiently small step size, though the required steps are impractically small. A follow-up work (Li et al., 2019) improved this by using the Moments Accountant (Abadi et al., 2016) to account for repeated data use under subsampled Gaussian mechanisms, enabling more realistic step sizes.

Other private posterior samplers have also been studied. Zhang & Zhang (2023) introduced a DP variant of the Metropolis–Hastings (MH) algorithm, and Räisä et al. (2021) proposed a DP Hamiltonian Monte Carlo (HMC) method. These rely on proposal distributions or momentum variables not present in diffusion sampling, and thus do not transfer directly to our setting. More recently, Bertazzi et al. (2025) proposed an analysis showing that privacy loss in SGLD can be concentrated in the final step by combining Girsanov's theorem with a perturbation trick. Their framework, however, requires strongly convex and Lipschitz-regularized objectives, which do not hold in our empirical denoiser formulation. Extending such ideas to diffusion sampling remains an intriguing future direction.

## 6 EXPERIMENTS

In this section, we provide a comprehensive privacy analysis of diffusion-based sampling under the $f$-DP composition framework. We examine the following critical aspects: (i) privacy guarantees for subsampled versus full training data, (ii) the role of critical windows in privacy loss accumulation and utility, and (iii) an approach to reduce privacy costs through public neural denoisers.

**Implementation**   We implement our modified reverse denoising process in PyTorch Paszke et al. (2019) and categorize our experiments into two regimes: (i) **full-batch**, where the entire dataset is used in each denoising step, and (ii) **mini-batch** subsampling, where uniformly sampled subsets of data are used at each iteration.

For noise scheduling, we adopt the exponential decay strategy of (Karras et al., 2022), where both the timesteps $t$ and their increment $\Delta t$ decrease exponentially according to a set of scheduling hyperparameters. The scheduler is parameterized by $\sigma_{\min}$, the smallest noise scale, $\sigma_{\max}$, the largest

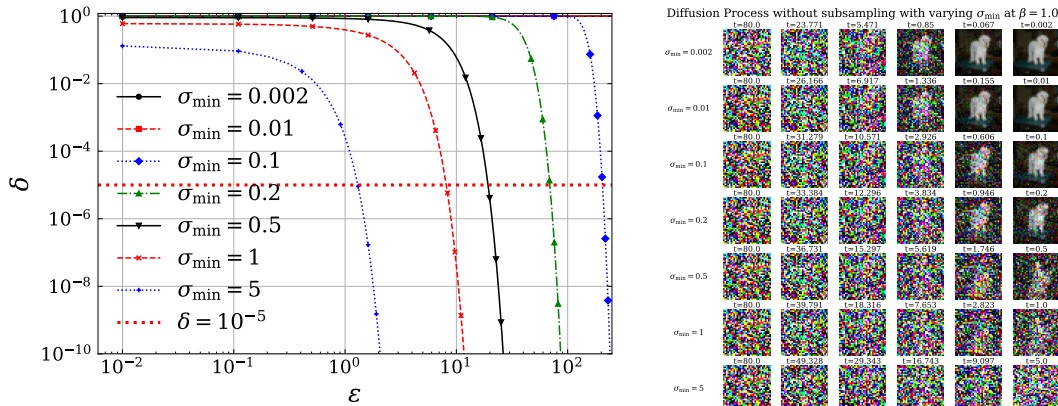

Figure 5: GDP Analysis of Diffusion-Based Sampling without Subsampling (full batch) and Corresponding Diffusion Process Visualization. The noise scheduling constants are $\sigma_{max} = 80$, $N = 50$, $\beta = 1$, and $\rho = 7$. **Left.** GDP conversion to $(\epsilon, \delta)$-DP for varying $\sigma_{\min}$. **Right.** Visualization of Diffusion Process with Varying $\sigma_{\min}$. Only 6 timesteps are shown for figure clarity.

noise scale, $\rho$, the noise decay rate, and $N$, the number of denoising steps. We experiment with $\sigma_{min} = 0.002, \sigma_{max} = 80, \rho = 7, N = 50$ otherwise noted. We also fix $p(\mathbf{x}_t)$ to a constant $\beta = 1$ for simplicity.

**Privacy Analysis**  Given the noise schedule and sensitivity, we calculate the corresponding $\mu$-GDP for each diffusion step (3). For the full-batch case, we derive the overall $\mu$-GDP using the $n$-fold composition of $\mu_i$-GDP mechanisms. For mini-batch subsampling, we calculate the asymptotic bound using the GDP central limit theorem (CLT) (Dong et al. (2022)), which provides tighter privacy guarantees due to subsampling amplification. This involves computationally calculating the $KL$-divergence and $\kappa_2$ (1).

**Datasets and Evaluation**  We evaluate our approach on 50K CIFAR-10 (Krizhevsky et al., 2009) training data, with ImageNet $32 \times 32$ (Russakovsky et al., 2015) used as a different source of public training data for the neural denoiser. Both $\mu$-GDP and $\mu$-GDP CLT are assessed through: (i) conversion to $(\epsilon, \delta)$-DP bounds and comparison of $\epsilon$ at a fixed $\delta = 10^{-5}$, and (ii) visualization of type I and type II error trade-off curves. We convert all $\mu$-GDP to $(\epsilon, \delta)$-DP using the closed-form solution: $\delta(\varepsilon) = \Phi\left(-\frac{\varepsilon}{\mu} + \frac{\mu}{2}\right) - e^{\varepsilon}\Phi\left(-\frac{\varepsilon}{\mu} - \frac{\mu}{2}\right)$, where $\Phi$ denotes the Gaussian cumulative distribution function.

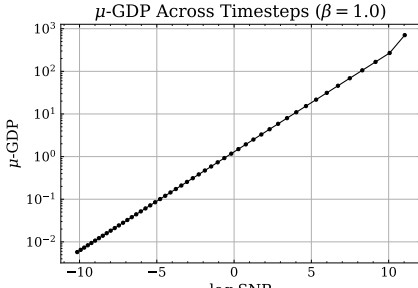

Figure 4: $\mu$-GDP for each time step with full batch of data.

### 6.1 PRIVACY ANALYSIS OF DIFFUSION-BASED SAMPLING WITH FULL TRAINING DATA

Our privacy analysis employs GDP composition theory to derive $(\epsilon, \delta)$-DP bounds for the diffusion sampling process. We vary the minimum noise level $\sigma_{\min}$ to understand its impact on privacy guarantees. Since CIFAR-10 comes with 50,000 training samples, we set delta to $10^{-5}$. Figure 4 shows how $\mu$ grows.

**Lower $\sigma_{\min}$ corresponds to less privacy**  Figure 5 demonstrates how the choice of $\sigma_{\min}$ fundamentally affects the privacy-utility tradeoff. As the $\sigma_{\min}$ decreases (getting closer to the training data), the GDP analysis suggests a worse $(\epsilon, \delta)$-DP guarantees.

**Full-batch empirical denoiser does not provide good utility for the privacy**  At $\epsilon \approx 1$ (corresponding to $\sigma_{min} = 5$), the generated sample at $t = 0$ is hard to distinguish from a pure noise

Table 1: Comparison of different DP generative models on CIFAR-10. Our method consists of sampling from an ImageNet-pretrained model to % of the generation process(Deng et al. (2009); Karras et al. (2022)). Afterwards, we switch to our empirical denoiser and sample for one denoising step.

| Method | $\epsilon$ | FID |
|---|---|---|
| DP-Diffusion (Ghalebikesabi et al. (2023)) | 10 | 9.8 |
| DP-LDM (Rombach et al. (2022b)) | 10 | 8.4 |
| DP-API (Lin et al. (2023)) | 0.67 | 7.87 |
| DP-MEPF (Harder et al. (2023)) | 10 | 29.1 |
| DP-DM (Dockhorn et al. (2023)) | 10 | 97.7 |
| DP-Promise (Wang et al. (2024)) | 10 | 17.9 |
| Ours ($p = 2 \times 10^{-5}$, 93% generation) | 3.848 | 2.912 |

by human eyes (bottom row in Figure 5-(b)). For a humanly discernable sample, we needed to fix $\sigma_{min} = 0.2$, which corresponds to $\epsilon \approx 90$.

## 6.2 PRIVACY ANALYSIS OF DIFFUSION-BASED SAMPLING WITH SUBSAMPLED DATA

As shown in Figure 6, the diffusion sampling with subsampled data significantly improves the full-batch sampling in terms of the type I and type II error profile. Individual profiles under each composition are shown in Figure 8. Figure 9 shows that subsampling boosts the quality of generated images. As $p$ (sampling rate) goes reduces the corresponding $\epsilon$ level also goes down (better privacy). However, we do not necessarily see the humanly discernable samples earlier in the denoising process with smaller $p$. This is due to the fact that with subsampling the empirical denoiser's accuracy also hurts.

To contextualize the effect of our privacy mechanisms, we report FID (Heusel et al., 2017)for a range of models, including our proposed *hybrid* pipelines that apply a neural denoiser in the early timesteps followed by an empirical denoiser in the later timesteps, in order to quantify how different denoising configurations impact utility under privacy noise. The results against baseline DP generative models are listed in Table 1, and we provide the full set of hybrid configurations in Appendix D.

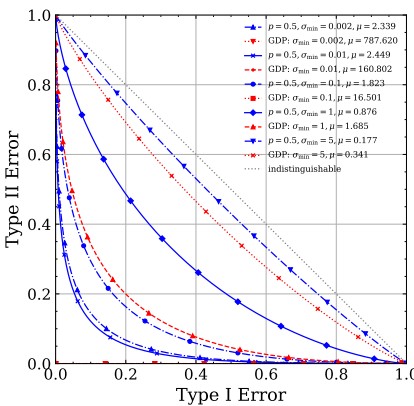

Figure 6: Type I and Type II Error under GDP and GDP CLT analyses with varying $\sigma_{min}$ and subsampling rate $p$. Notice the large difference between two $\mu$ values when full batch was used ($\mu \approx 787$) and subsampled at $p = 0.5$ was used ($\mu \approx 2.3$), when $\sigma_{min} = 0.002$.

## 6.3 PRIVACY IN RELATION TO THE CRITICAL WINDOWS

We extend our privacy analyses to quantify privacy levels based on where we stop relative to the critical window of the denoising process. Georgiev et al. (2023) empirically show the existence of a critical window, and depending on where we stop in relation to the critical window, our privacy loss will differ. In Figure 10, we show how the same $(\epsilon, \delta)$-DP guarantees and Type I and II error profiles change depending on where we stop denoising. Early stopping is sensible due to the existence of critical windows. But it is also practical for reducing privacy losses by limiting the number of denoising steps using private data.

Empirically, we can detect the range of the critical window using a pre-trained zero-shot CLIP model as a classifier (Radford et al., 2021), following (Georgiev et al., 2023). We start our denoising process from $t = T$ and continue until we stop at several time points around where the critical window emerges. When $t < 0.1$, the probability of an image being generated belonging to the *automobile* class is near 1, as shown in Fig. 7. We pick three points around where the critical window emerges: the probability is nearly 1 at $t = 0.05$, close to 0.7 at $t = 0.09$, and around 0.1 at $t = 0.5$. If we stop early in relation to where the critical point emerges (i.e., before the probability soaring to 1), we get a better value of $\mu$, but the sample looks very noisy. However, if we stop later than where the critical point emerges, we get a larger value of $\mu$ but the sample quality becomes better..

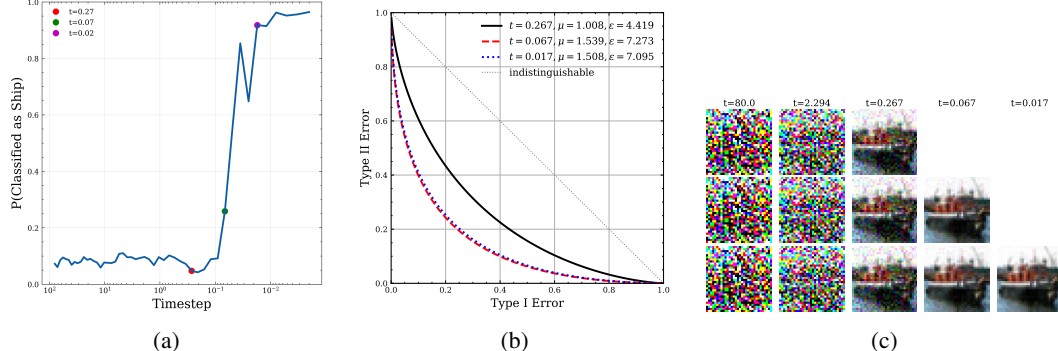

(a)             (b)             (c)

Figure 7: Detection of critical window. **(a)** zero-shot CLIP model detects the probability of an image being generated belonging to the *automobile* class. **(b)** Privacy costs at three time points. Note that the blue trace overlaps with the x-axis. **(c)** Generated images denoised at varying timesteps.

Table 2: Membership inference attacks (MIA): our theoretical ASR closely matches that of the numerical ASR, as $t$ gets smaller, hinting the privacy analysis of the empirical denoiser is comparable to that of the neural denoiser.

| $t$ | $T$ | $\frac{3}{4}T$ | $\frac{1}{2}T$ | $\frac{1}{4}T$ |
|---|---|---|---|---|
| cosine similarity shown in Fig. 1 | 1.000 | 0.999 | 0.956 | 0.845 |
| theoretical ASR at $p = 0.5$ | 0.5019 | 0.5204 | 0.6909 | 0.9999 |
| theoretical ASR at $p = 0.01$ | 0.5000 | 0.5002 | 0.5031 | 0.6927 |
| theoretical ASR at $p = 0.005$ | 0.5000 | 0.5001 | 0.5012 | 0.6371 |
| numerical ARS | 0.5005 | 0.5001 | 0.5072 | 0.5096 |

## 6.4 PRIVACY IMPLICATIONS OF NEURAL AND EMPIRICAL DENOISERS

We numerically evaluate the privacy implications of these two denoisers by measuring their attack success rates (ASR) in membership inference attacks (MIAs). Using the MIA method in (Duan et al., 2023b), Table 2 shows that our theoretical ASR (See Sec. G for details) based on $\epsilon$ guarantees on our empirical denoiser closely matches that of the numerical ASR based on the neural denoiser trained with CIFAR10 data, as $t$ gets smaller. The theoretical ASR is calculated at the subsampling rates $p = 0.5, 0.01, 0.005$. We used the test data of the CIFAR-10 dataset as a held-out set for attacks.

Regarding the effectiveness of MIAs, Duan et al. (2023b) show that the performance of MIA remains high across $t \in [250, 50]$ in the DDPM timesteps. The difference between the neural and empirical denoisers in this range is still relatively small, as shown in Fig. 1 (Cosine similarity above 0.8).

## 7 CONCLUSION AND DISCUSSION

We provide the first systematic privacy analysis of diffusion-based sampling. By introducing an empirical denoiser and framing each denoising step as a Gaussian mechanism, we applied Gaussian Differential Privacy (GDP) to quantify how privacy loss accumulates during sampling. Our results show that leakage is highly non-uniform across timesteps, concentrating in critical windows where semantic structure emerges, while subsampling significantly amplifies privacy guarantees. These findings suggest that privacy in diffusion models is not solely determined by training but can be shaped by sampling strategies such as early stopping and hybrid use of public denoisers. In this way, our framework bridges the gap between practical generative pipelines and formal privacy guarantees, opening the door to scalable and privacy-aware diffusion sampling. While our analysis with Gaussian noise is not straightwardedly applicable to the discrete diffusion models, it would be an intriguing future direction on how to interpret the discrete transition kernel as a DP mechanism, applicable to text-diffusion models.

## ETHICS STATEMENT

Our work studies the privacy implications of diffusion-based sampling. The motivation is to provide a principled framework for understanding and mitigating privacy risks in generative models, including applications in sensitive domains such as human face data or biological gene expressions. By offering formal guarantees through differential privacy (DP), our analysis aims to reduce the likelihood that individual training examples can be memorized or extracted. We note, however, that DP provides quantifiable but not absolute protection, and our results should be interpreted in this light.

This work does not involve human subjects or sensitive personal data. All experiments are conducted on publicly available datasets commonly used in generative modeling. We encourage future applications of our methods to carefully consider domain-specific ethical implications, particularly in high-stakes settings such as healthcare or finance. Ultimately, our goal is to advance the responsible development of privacy-preserving generative models.

## REPRODUCIBILITY STATEMENT

We have taken several steps to ensure the reproducibility of our results. All theoretical results are stated with explicit assumptions. The Gaussian Differential Privacy composition bounds are fully detailed in Sections 3 and 4. For empirical results, we describe all details in the main text and appendix, using only publicly available benchmarks commonly employed in generative modeling. To further facilitate reproducibility, we will release the source code after acceptance.

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

# A  HOW TO USE THE CLT GDP ANALYSIS IN OUR CASE

Theorem 3.5 in (Dong et al., 2022) states that the compsition of many $f$-DP mechanisms is asyptotically $2K/s$-GDP. The original theorem is stated below:

**Theorem 1.** *[Theorem 3.5 in (Dong et al., 2022)] Let $\{f_{ni} : 1 \le i \le n\}_{n=1}^{\infty}$ be a triangular array of symmetric trade-off functions and assume the following limits for some constants $K \ge 0$ and $s > 0$ as $n \to \infty$:*

1.  $\sum_{i=1}^{n} \mathrm{kl}(f_{ni}) \to K$;

2.  $\max_{1 \le i \le n} \mathrm{kl}(f_{ni}) \to 0$;

3.  $\sum_{i=1}^{n} \kappa_2(f_{ni}) \to s^2$;

4.  $\sum_{i=1}^{n} \kappa_3(f_{ni}) \to 0$.

*Then, we have*

$$\lim_{n \to \infty} f_{n1} \otimes f_{n2} \otimes \cdots \otimes f_{nn}(\alpha) = G_{2K/s}(\alpha)$$

*uniformly for all $\alpha \in [0, 1]$.*

Now, we have to calculate $kl$ and $\kappa_2$ functions to be able to compute the final $\mu$-GDP level. In the case of subsampled Gaussian mechanisms with varying noise scales, we use the definition of $kl$ and $\kappa_2$ from Lemma 5.5 in (Dong et al., 2022). The original lemma is stated below.

**Lemma A.1** (Lemma 5.5 in (Dong et al., 2022))**.** Let $Z(x) = \log\left(p \cdot e^{\mu x - \mu^2/2} + 1 - p\right)$ and $\varphi(x) = \frac{1}{\sqrt{2\pi}} e^{-x^2/2}$ be the density of the standard normal distribution. Then

$$\mathrm{kl}(C_p(G_\mu)) = p \int_{\mu/2}^{+\infty} Z(x) \cdot \big(\varphi(x - \mu) - \varphi(x)\big) \, dx,$$

$$\kappa_2(C_p(G_\mu)) = \int_{\mu/2}^{+\infty} Z^2(x) \cdot \big(p\varphi(x - \mu) + (2 - p)\varphi(x)\big) \, dx,$$

$$\bar{\kappa}_3(C_p(G_\mu)) = \int_{\mu/2}^{+\infty} \big|Z(x) - \mathrm{kl}(C_p(G_\mu))\big|^3 \cdot \big(p\varphi(x - \mu) + (1 - p)\varphi(x)\big) \, dx$$

$$+ \int_{\mu/2}^{+\infty} \big|Z(x) + \mathrm{kl}(C_p(G_\mu))\big|^3 \cdot \varphi(x) \, dx.$$

Note that $C_p(G_\mu)$ is the trade-off function of a p-subsampled $\mu$-GDP mechanism. Following the definition in eq. 14 in (Dong et al., 2022) given as: For a symmetric trade-off function $f$ and a unique fixed point of f denoted by $x^*$, that is, $f(x^*) = x^*$, we have

$$C_p(f) = \begin{cases} f_p(x), & x \in [0, x^*] \\ x^* + f_p(x^*) - x, & x \in [x^*, f_p(x^*)] \\ f_p(x)^{-1}, & x \in [f_p(x^*), 1]. \end{cases} \tag{6}$$

We simply replace $f$ in eq. 6 with $G_\mu$ to define $C_p(G_\mu)$. In our case, each Gaussian mechanism takes a different $\sigma_t$, so we need to numerically evaluate $\mathrm{kl}(C_p(G_{\frac{1}{\sigma_t}}))$ and $\kappa_2(C_p(G_{\frac{1}{\sigma_t}}))$ for each $\sigma_t$. Then, we sum them up to compute $K$ and $s$ to finally obtain $2K/s$, which is the final $\mu$-GDP level.

# B  DIFFERENTIAL PRIVACY AND GAUSSIAN DIFFERENTIAL PRIVACY

## B.1  DIFFERENTIAL PRIVACY

The concept of differential privacy (DP) (Abadi et al., 2016) provides a formal guarantee that a model behaves almost identically regardless of whether any particular data sample is included in the training set. In other words, DP bounds the extent to which a single sample can influence the model's

behavior. More precisely, a mechanism $\mathcal{M}$, , which may be a training algorithm such as Stochastic Gradient Descent (SGD) or, more generally, any randomized algorithm including diffusion sampling, is said to be $\epsilon$-DP if the privacy loss is bounded by $\epsilon$: $\log \frac{Pr[\mathcal{M}(\mathcal{D})=o]}{Pr[\mathcal{M}(\mathcal{D}')=o]} \leq \epsilon$, where $\mathcal{D}$ and $\mathcal{D}'$ differ in the record of a single individual; and $\mathcal{M}(\mathcal{D}) = o$ denotes that applying the mechanism $\mathcal{M}$ on a dataset $\mathcal{D}$ ends up converging to the output event $o$.

For instance, if $\mathcal{M}$ is a training optimizer such as SGD, then $\mathcal{M}(\mathcal{D})$ is the trained model and $Pr(\mathcal{M}(\mathcal{D}) = o)$ depicts the probability the model $o$ being the trained one. Intuitively, if deleting a single data point in $\mathcal{D}$ to obtain $\mathcal{D}'$ does not substantially change the probability of obtaining the same output model $o$, then the training algorithm can be regarded as privacy-preserving[1]. Put more simply, if the probability the model converges to the same parameters is relatively equivalent regardless of whether a particular sample is present, then that sample cannot be said to meaningfully affect training. Extending this definition, if the mechanism $\mathcal{M}$ satisfies the bound with probability at least $1 - \delta$, i.e.,

$$Pr\left[\log \frac{Pr[\mathcal{M}(\mathcal{D}) = o]}{Pr[\mathcal{M}(\mathcal{D}') = o]} \geq \epsilon\right] \leq \delta,$$

then the mechanism is called $(\epsilon, \delta)$-DP.

A natural building block for designing private (either training or sampling) algorithms is the *Gaussian mechanism*. This is particularly relevant for our setting, since each step in diffusion sampling injects Gaussian noise. Formally, given a function $\mathbf{h} : \mathcal{D} \mapsto \mathbb{R}^p$, the *Gaussian mechanism* releases

$$\mathcal{M}_{\text{Gauss}}(\mathcal{D}; \mathbf{h}) = \mathbf{h}(\mathcal{D}) + \boldsymbol{n}, . \tag{7}$$

where $\boldsymbol{n} \sim \mathcal{N}(0, \sigma^2 \Delta_{\mathbf{h}}^2 \mathbf{I})$. Here, $\Delta_{\mathbf{h}}$ denotes the *global sensitivity*—the maximum $L_2$ difference between outputs on two neighboring datasets—and $\sigma$ controls the noise level as a function of $(\epsilon, \delta)$. By adding noise calibrated to sensitivity, the Gaussian mechanism ensures that the presence or absence of any single sample only has a limited influence on the output. This mechanism underlies differentially private training methods such as DP-SGD (Abadi et al., 2016), and, as we will show, it also directly connects to the diffusion sampling process we analyze in this work.

As an interesting fact in differential privacy, it is widely known that subsampling amplifies privacy (Kasiviswanathan et al., 2011). For example, if we flip a coin to decide whether each entry of the training data is in or out to form a smaller dataset, we would expect the mechanism using the smaller dataset to be twice as private as the original mechanism using the full dataset. With a flipping probability of $50\%$, every individual benefits from perfect privacy if the individual is not included in the smaller dataset. To benefit from the privacy amplification effect, a subsampled Gaussian mechanism (applying the Gaussian mechanism on randomly subsampled data) is also widely used (Abadi et al., 2016; Wang et al., 2019). We will also study the effect of subsampling on the privacy loss incurring during the diffusion sampling process.

### B.2 GAUSSIAN DIFFERENTIAL PRIVACY

While the Gaussian mechanism provides a natural foundation for analyzing privacy, the classical $(\epsilon, \delta)$-DP framework yields only loose bounds when these mechanisms are composed, often resulting in substantially looser guarantees than the actual privacy loss. To address this, we adopt the more refined $f$-DP framework (Dong et al., 2022), which characterizes privacy through hypothesis testing and provides an exact trade-off curve (function), rather than the looser worst-case bounds of $(\epsilon, \delta)$-DP.

A particularly important special case is *Gaussian Differential Privacy (GDP)*, which describes the privacy profile of the Gaussian mechanism exactly using a single parameter $\mu$. For example, in the case of eq. 7 with $\boldsymbol{n} \sim \mathcal{N}(0, \sigma^2 \Delta_{\mathbf{h}}^2 \mathbf{I})$, the Gaussian mechanism corresponds to $\frac{1}{\sigma}$-GDP. For a Gaussian mechanism, we can define a trade-off function $T$ using the standard normal CDF $\Phi$ given by,

$$G_\mu := T(\mathcal{N}(0, 1), \mathcal{N}(\mu, 1). \quad G_\mu(\alpha) := \Phi(\Phi^{-1}(1 - \alpha) - \mu). \tag{8}$$

---

[1]SGD and Adam are not DP algorithms as the gradient update may be dominated by a huge outlier that significantly affects the model's parameters. As a result, a model trained with SGD/Adam can potentially reveal information about such outliers, leaving the data vulnerable to extraction attacks.

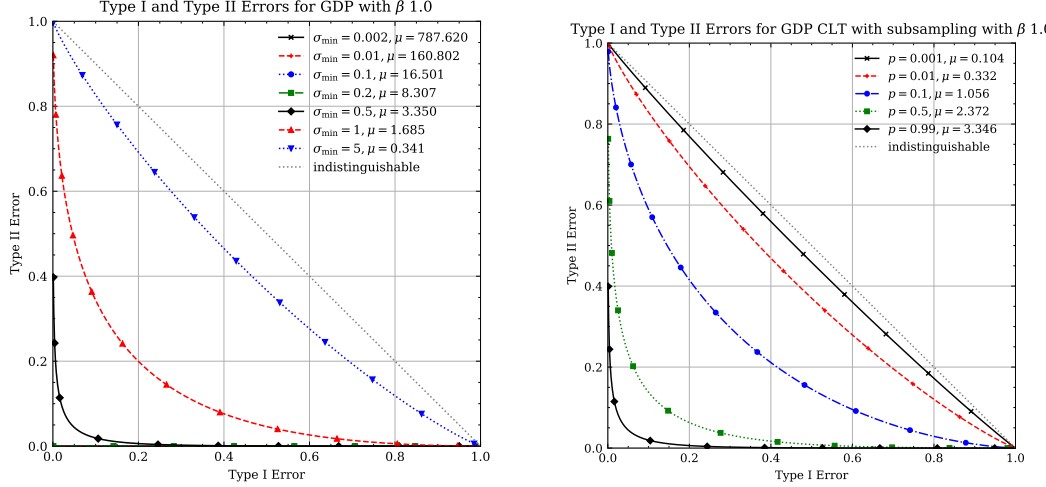

Figure 8: Type I and Type II Error under GDP and GDP CLT analyses with varying $\sigma_{min}$ and subsampling rate

This makes GDP especially well aligned with diffusion sampling, where Gaussian perturbations are intrinsic to every denoising step. Moreover, GDP is closed under composition (Corollary 2 of Dong et al. (2022)): the cumulative privacy loss of $n$ mechanisms with parameters $\mu_1, \ldots, \mu_n$ is given tightly by

$$\mu = \sqrt{\sum_{i=1}^{n} \mu_i^2}. \tag{9}$$

Thus, GDP not only matches the Gaussian mechanism exactly but also enables precise accounting of privacy loss across many Gaussian steps that constitute diffusion sampling.

For subsampled Gaussian mechanisms, the trade-off function in eq. 8 needs to be modified to incorporate the randomness coming from the subsampling process and the GDP composition in eq. 9 is no longer valid. In this case, one can benefit from using the central limit theorem, which tells us that composing many $f$-DP mechanisms converges to GDP (where the level of $\mu$ in the final GDP depends on several functions which we will discuss in Appendix), as formally stated in Theorem 3.5 of Dong et al. (2022).

## C MORE PLOTS

### C.1 TYPE I AND II ERRORS UNDER GDP AND GDP CLT ANALYSES

Fig. 8 shows the comparison between GDP and GDP CLT analyses in terms of Type I and II errors with varying $\sigma_{min}$ and subsampling rate $p$.

### C.2 PRIVACY LOSS IN RELATION TO CRITICAL WINDOWS

Fig. 10 shows $(\epsilon, \delta)$-DP traces (Top) and Type I and II error profiles (bottom) under GDP and GDP CLT with varying stopping time points in the reverse process.

## D EVALUATION OF OUR HYBRID EMPIRICAL DENOISER ON CIFAR10

We conducted a series of experiments to evaluate the hybrid denoising process. In particular, we analyze how switching from the neural denoiser to the empirical denoiser influences both privacy

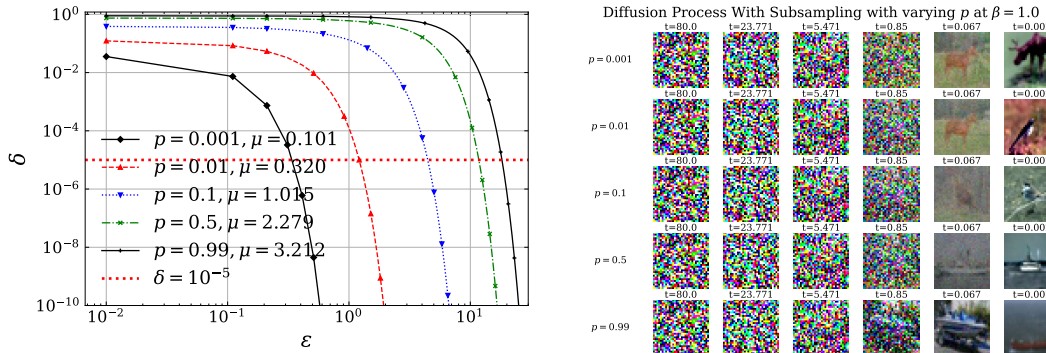

Figure 9: GDP CLT Analysis of Diffusion-Based Sampling with Subsampling and Corresponding Diffusion Process Visualization. The noise scheduling constants are $\sigma_{max} = 80$, $N = 50$, $\beta = 1$, $\rho = 7$, and $\sigma_{\min} = 0.01$. **Left.** GDP CLT conversion to $(\epsilon, \delta)$-DP for varying subsampling rate $p$. **Right.** Visualization of Diffusion Process with varying subsampling rate. Only 6 timesteps are shown for clarity.

guarantees and image quality. We vary three factors: the subsampling rate $p$, the point in the generation trajectory at which we switch to the empirical denoiser, and the number of denoising steps done by the empirical denoiser.

All experiments use an ImageNet-pretrained diffusion model for the neural denoiser, and we adopt the exponential decay strategy of EDM (Karras et al. (2022)). The scheduler is parameterized by the minimum and maximum noise levels $(\sigma_{min}, \sigma_{max})$, the noise decay rate $\rho$, and the number of denoising steps $N$. We experiment with $\sigma_{min} = 0.002, \sigma_{max} = 80, \rho = 7, N = 39$. We also fix $p(x_t)$ to a constant $\beta = 1$ for simplicity in the empirical denoiser. For both experiments, we examine the subsampling mini-batch size of $[1, 5, 10, 25, 50, 100]$, corresponding to the subsampling rate $p = [2 \times 10^{-5}, 1 \times 10^{-4}, 2 \times 10^{-4}, 5 \times 10^{-4}, 1 \times 10^{-3}, 2 \times 10^{-3}]$. We also experimented with different switching points, expressed as the percentage of generation completed by the neural denoiser.

In the first experiment, we analyzed the multi-step empirical denoiser. The empirical denoiser is applied repeatedly from the transition point to $\sigma_{min}$. We monitored the resulting $\epsilon$ and FID scores under varying subsampling rates, shown in Table 3. We observe that FID decreases as the subsampling rate increases, which corresponds to larger values of $\epsilon$. The best image quality is obtained with an FID of 111.669 at $\epsilon = 99.749$, using a subsampling rate of $p = 2 \times 10^{-3}$ and switching to the empirical denoiser after $85\%$ of the neural generation process.

In the second experiment, we examined the single-step empirical denoiser. Instead of multiple steps, the empirical denoiser is applied only once after a portion of the neural generation process. This single-step variant incurs substantially less privacy loss than the multi-step setting, as shown in Table 4. Empirically, we observe a critical window between $83\%$ and $93\%$ generation, where switching to the empirical denoiser produces high-quality samples at a relatively low privacy cost. The best image quality is achieved with an FID of 2.912 at $\epsilon = 3.848$, using a subsampling rate of $p = 2 \times 10^{-5}$ and switching at $93\%$ neural generation.

## E   MEMBERSHIP INFERENCE ATTACKS ON NEURAL DENOISERS AND EMPIRICAL DENOISERS

## F   DETECTION OF CRITICAL WINDOWS WITH ZERO-HOT CLIP MODELS

## G   COMPUTING THEORETICAL ASR

Following Triastcyn & Faltings (2020), we calculate the theoretical ARS as follows. Suppose an adversary observed an $\epsilon$-DP release $o$ and tried to distinguish between two candidate datasets

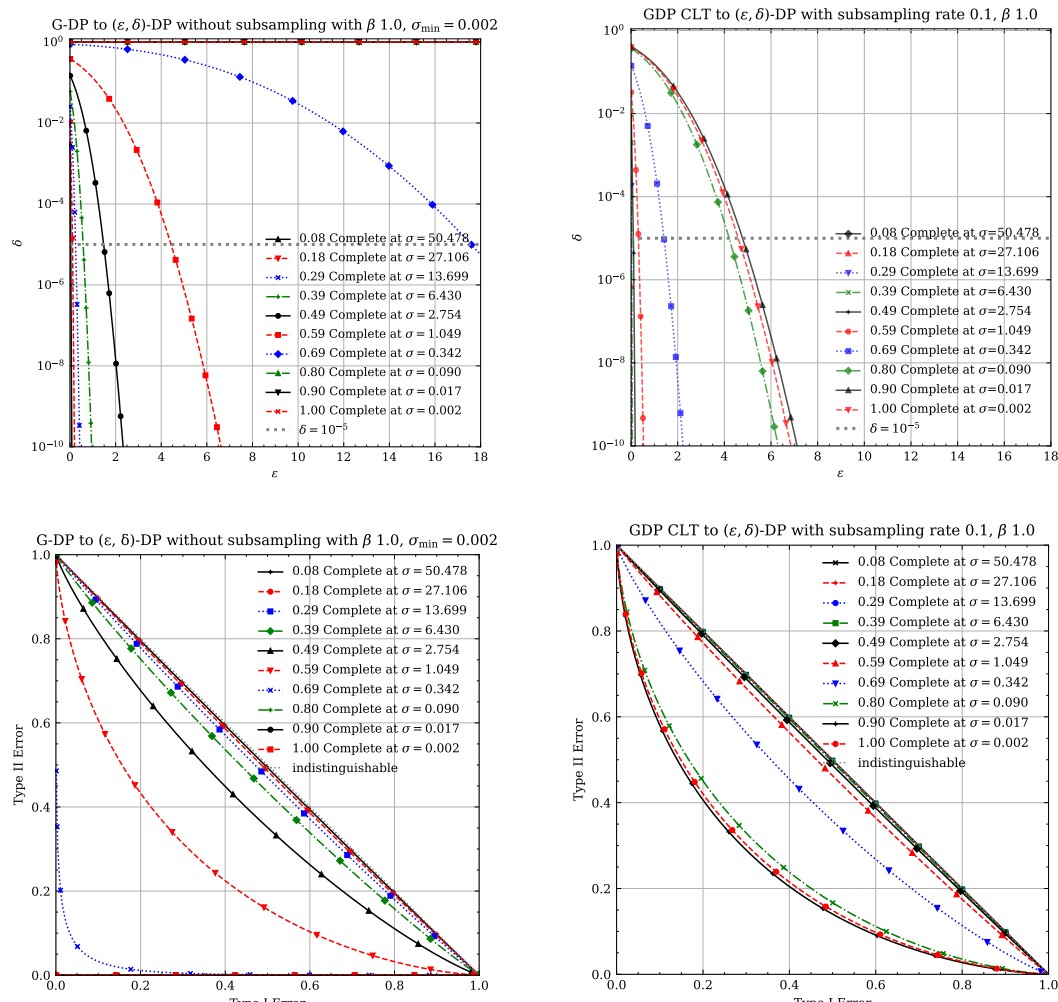

Figure 10: $(\epsilon, \delta)$-DP traces (Top) and Type I and II error profiles (Bottom) under GDP and GDP CLT with varying stopping time points in the reverse process

$\mathcal{D}, \mathcal{D}'$ with a uniform prior $p(\mathcal{D}) = p(\mathcal{D}') = 0.5$. Without loss of generality, let $\mathcal{D}$ be the correct dataset, then the adversary will infer the probability $p(\mathcal{D}|o)$ as $p(\mathcal{D}|o) = \frac{p(o|\mathcal{D})p(\mathcal{D})}{p(o|\mathcal{D})p(\mathcal{D}) + p(o|\mathcal{D}')p(\mathcal{D}')} \leq \frac{p(o|\mathcal{D})}{p(o|\mathcal{D}) + \exp(-\epsilon)p(o|\mathcal{D})} = \frac{1}{1 + \exp(-\epsilon)}$. So, for $\epsilon = 1$ the upperbound on the membership inference success probability is 73.1%, for $\epsilon = 5$, it is 99.33%, and for $\epsilon = 10$, it is 99.995%. Futhremore, the success probability of MIAs typically similar or lower than the theoretical success rate as shown in (Lowy et al., 2024).

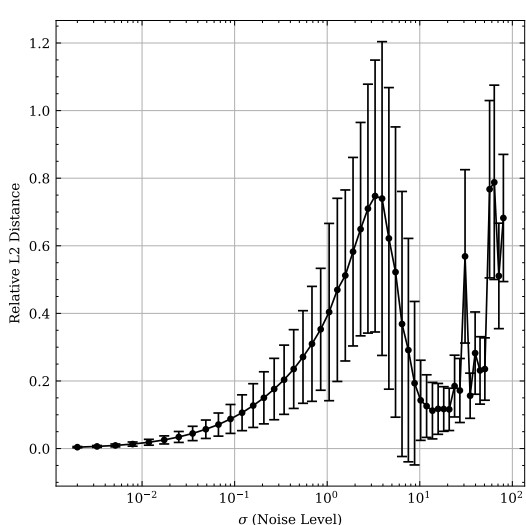

Figure 11: Relative distance between the denoising directions, i.e., $\frac{\|\hat{\mathbb{E}}-D\|}{D}$, by empirical and neural denoisers

Table 3: Results of multi-step denoising using the empirical denoiser to minimum sigma value $\sigma_{\min}$ after % of generation with neural denoiser, with sub-sampling rate $p$, privacy level $\epsilon$ with $\delta = 10^{-5}$, and sample quality (FID) for CIFAR10.

| % Generation | $\sigma_{\min}$ | $p$ | $\epsilon$ | FID |
|---|---|---|---|---|
| 95 | 0.002 | $2.00 \times 10^{-5}$ | 3.849 | 188.605 |
| 95 | 0.002 | $1.00 \times 10^{-4}$ | 9.997 | 183.108 |
| 95 | 0.002 | $2.00 \times 10^{-4}$ | 15.456 | 176.715 |
| 95 | 0.002 | $5.00 \times 10^{-4}$ | 28.373 | 167.757 |
| 95 | 0.002 | $1.00 \times 10^{-3}$ | 46.211 | 160.660 |
| 95 | 0.002 | $2.00 \times 10^{-3}$ | 77.330 | 155.318 |
| 93 | 0.002 | $2.00 \times 10^{-5}$ | 4.401 | 179.445 |
| 93 | 0.002 | $1.00 \times 10^{-4}$ | 11.547 | 167.333 |
| 93 | 0.002 | $2.00 \times 10^{-4}$ | 17.966 | 155.722 |
| 93 | 0.002 | $5.00 \times 10^{-4}$ | 33.321 | 139.432 |
| 93 | 0.002 | $1.00 \times 10^{-3}$ | 54.753 | 128.167 |
| 93 | 0.002 | $2.00 \times 10^{-3}$ | 99.932 | 118.684 |
| 90 | 0.002 | $2.00 \times 10^{-5}$ | 4.576 | 179.443 |
| 90 | 0.002 | $1.00 \times 10^{-4}$ | 12.045 | 168.128 |
| 90 | 0.002 | $2.00 \times 10^{-4}$ | 18.776 | 153.380 |
| 90 | 0.002 | $5.00 \times 10^{-4}$ | 34.929 | 136.057 |
| 90 | 0.002 | $1.00 \times 10^{-3}$ | 57.547 | 125.364 |
| 90 | 0.002 | $2.00 \times 10^{-3}$ | 97.474 | 117.334 |
| 88 | 0.002 | $2.00 \times 10^{-5}$ | 4.635 | 178.597 |
| 88 | 0.002 | $1.00 \times 10^{-4}$ | 12.906 | 166.409 |
| 88 | 0.002 | $2.00 \times 10^{-4}$ | 19.142 | 152.637 |
| 88 | 0.002 | $5.00 \times 10^{-4}$ | 35.460 | 136.455 |
| 88 | 0.002 | $1.00 \times 10^{-3}$ | 58.819 | 123.697 |
| 88 | 0.002 | $2.00 \times 10^{-3}$ | 99.152 | 112.168 |
| 85 | 0.002 | $2.00 \times 10^{-5}$ | 4.655 | 179.466 |
| 85 | 0.002 | $1.00 \times 10^{-4}$ | 12.269 | 165.492 |
| 85 | 0.002 | $2.00 \times 10^{-4}$ | 19.142 | 154.461 |
| 85 | 0.002 | $5.00 \times 10^{-4}$ | 35.660 | 136.985 |
| 85 | 0.002 | $1.00 \times 10^{-3}$ | 58.819 | 123.697 |
| 85 | 0.002 | $2.00 \times 10^{-3}$ | 99.749 | 111.669 |
| 83 | 0.002 | $2.00 \times 10^{-5}$ | 4.663 | 179.722 |
| 83 | 0.002 | $1.00 \times 10^{-4}$ | 12.291 | 166.993 |
| 83 | 0.002 | $2.00 \times 10^{-4}$ | 19.178 | 153.264 |
| 83 | 0.002 | $5.00 \times 10^{-4}$ | 35.732 | 137.327 |
| 83 | 0.002 | $1.00 \times 10^{-3}$ | 58.944 | 123.819 |
| 83 | 0.002 | $2.00 \times 10^{-3}$ | 99.973 | 112.394 |
| 75 | 0.002 | $2.00 \times 10^{-5}$ | 4.667 | 179.096 |
| 75 | 0.002 | $1.00 \times 10^{-4}$ | 12.304 | 166.502 |
| 75 | 0.002 | $2.00 \times 10^{-4}$ | 19.178 | 150.659 |
| 75 | 0.002 | $5.00 \times 10^{-4}$ | 35.775 | 137.784 |
| 75 | 0.002 | $1.00 \times 10^{-3}$ | 59.020 | 125.472 |
| 75 | 0.002 | $2.00 \times 10^{-3}$ | $> 1000$ | 112.683 |
| 50 | 0.002 | $2.00 \times 10^{-5}$ | 4.668 | 179.145 |
| 50 | 0.002 | $1.00 \times 10^{-4}$ | 12.306 | 165.949 |
| 50 | 0.002 | $2.00 \times 10^{-4}$ | 19.178 | 155.753 |
| 50 | 0.002 | $5.00 \times 10^{-4}$ | 35.777 | 139.381 |
| 50 | 0.002 | $1.00 \times 10^{-3}$ | 59.023 | 126.682 |
| 50 | 0.002 | $2.00 \times 10^{-3}$ | $> 1000$ | 116.090 |

Table 4: Results of single-step denoising using the empirical denoiser after % of generation with neural denoiser, with sub-sampling rate $p$, privacy level $\epsilon$ with $\delta = 10^{-5}$, and sample quality (FID) for CIFAR10.

| $p$ | % Generation | $\epsilon$ | FID |
|---|---|---|---|
| $2.00 \times 10^{-5}$ | 95 | 3.849 | 188.605 |
| $1.00 \times 10^{-4}$ | 95 | 9.997 | 183.108 |
| $2.00 \times 10^{-4}$ | 95 | 15.456 | 176.715 |
| $5.00 \times 10^{-4}$ | 95 | 28.373 | 167.757 |
| $1.00 \times 10^{-3}$ | 95 | 46.211 | 160.660 |
| $2.00 \times 10^{-3}$ | 95 | 77.330 | 155.318 |
| $2.00 \times 10^{-5}$ | 93 | 3.848 | **2.912** |
| $1.00 \times 10^{-4}$ | 93 | 9.997 | 7.325 |
| $2.00 \times 10^{-4}$ | 93 | 15.456 | 10.786 |
| $5.00 \times 10^{-4}$ | 93 | 28.372 | 14.320 |
| $1.00 \times 10^{-3}$ | 93 | 46.209 | 16.657 |
| $2.00 \times 10^{-3}$ | 93 | 77.327 | 18.842 |
| $2.00 \times 10^{-5}$ | 90 | 3.848 | 4.929 |
| $1.00 \times 10^{-4}$ | 90 | 9.996 | 10.346 |
| $2.00 \times 10^{-4}$ | 90 | 15.454 | 13.810 |
| $5.00 \times 10^{-4}$ | 90 | 28.370 | 17.120 |
| $1.00 \times 10^{-3}$ | 90 | 46.204 | 19.811 |
| $2.00 \times 10^{-3}$ | 90 | 77.318 | 21.576 |
| $2.00 \times 10^{-5}$ | 88 | 3.847 | 9.317 |
| $1.00 \times 10^{-4}$ | 88 | 9.993 | 13.092 |
| $2.00 \times 10^{-4}$ | 88 | 15.450 | 19.162 |
| $5.00 \times 10^{-4}$ | 88 | 28.361 | 23.529 |
| $1.00 \times 10^{-3}$ | 88 | 46.191 | 26.593 |
| $2.00 \times 10^{-3}$ | 88 | 77.294 | 28.650 |
| $2.00 \times 10^{-5}$ | 85 | 3.845 | 17.991 |
| $1.00 \times 10^{-4}$ | 85 | 9.986 | 25.608 |
| $2.00 \times 10^{-4}$ | 85 | 15.439 | 30.489 |
| $5.00 \times 10^{-4}$ | 85 | 28.339 | 35.714 |
| $1.00 \times 10^{-3}$ | 85 | 46.152 | 38.427 |
| $2.00 \times 10^{-3}$ | 85 | 77.226 | 40.877 |
| $2.00 \times 10^{-5}$ | 83 | 3.838 | 34.508 |
| $1.00 \times 10^{-4}$ | 83 | 9.968 | 44.141 |
| $2.00 \times 10^{-4}$ | 83 | 15.409 | 48.749 |
| $5.00 \times 10^{-4}$ | 83 | 28.281 | 54.302 |
| $1.00 \times 10^{-3}$ | 83 | 46.053 | 58.895 |
| $2.00 \times 10^{-3}$ | 83 | 77.051 | 61.948 |
| $2.00 \times 10^{-5}$ | 75 | 3.576 | 123.473 |
| $1.00 \times 10^{-4}$ | 75 | 9.331 | 137.552 |
| $2.00 \times 10^{-4}$ | 75 | 14.439 | 141.587 |
| $5.00 \times 10^{-4}$ | 75 | 26.517 | 147.779 |
| $1.00 \times 10^{-3}$ | 75 | 43.182 | 150.616 |
| $2.00 \times 10^{-3}$ | 75 | 72.241 | 152.010 |
| $2.00 \times 10^{-5}$ | 50 | 0.001 | 431.166 |
| $1.00 \times 10^{-4}$ | 50 | 0.009 | 431.501 |
| $2.00 \times 10^{-4}$ | 50 | 0.020 | 432.066 |
| $5.00 \times 10^{-4}$ | 50 | 0.057 | 431.402 |
| $1.00 \times 10^{-3}$ | 50 | 0.121 | 431.332 |
| $2.00 \times 10^{-3}$ | 50 | 0.258 | 431.645 |

