# OpenReview forum: "How private is diffusion-based sampling?"
_ICLR.cc/2026/Conference — Submitted to ICLR 2026_

### Official Review · Reviewer_KyYx · 2025-10-28

**Soundness:** 1
**Presentation:** 1
**Contribution:** 1
**Rating:** 2
**Confidence:** 3

**Summary:**

This paper studies privacy leakage during the sampling process of diffusion models. It replaces the intractable neural denoiser with an empirical denoiser by dataset average so that each reverse step can be written as a Gaussian mechanism with sensitivity controlled by clipping. This allows per-step \mu-GDP accounting and composition across timesteps. The paper also argues that privacy loss concentrates in “critical windows” when semantics emerge; it further suggests a hybrid pipeline that switches to a public (non-private) denoiser outside those windows.

**Strengths:**

- The “critical window” framing offers a coherent qualitative perspective on where privacy loss concentrates along the sampling trajectory.
- A GDP-based accounting is presented, leveraging the exact $\mu$-composition property for Gaussian mechanisms.

**Weaknesses:**

1. **Lack of experimental baselines and validation.**
    - No comparison to other DP methods.
    - Image quality is not measured (the paper explicitly avoids FID/IS), so practical impact on generation quality is unclear.

2. **“Critical window” claims lack an operational detector.**
    - The idea is qualitative only; no quantitative rule (e.g., change-point in $\mu_t$, SNR threshold, or semantic-classifier stability) is provided or evaluated.

3. **Clarity and notation issues.**
    - Line ~163: What is $\Delta$?
    - Lines ~185–186: What is $C$? Is this the clip norm used to bound sensitivity?
    - Line ~201: What is $\mu_{t_i}$ and why is it defined that way?
    - Line 460: “Figure 3” is referenced but not linked/connected.

**Questions:**

- Can you provide baselines against DP-trained diffusion (e.g., DP-SGD) at matched privacy budgets, and report FID/IS (or CLIP-based metrics) to quantify utility.
- Can you provide a quantitative detector for the “critical window” (e.g., a change-point on per-step or cumulative \mu, an SNR threshold, or a classifier-stability metric), with ablations?

---

> ### Author Response · Authors · 2025-11-22
>
> Thanks for excellent suggestions on the FID scores and the critical window detection.
>
> ## Q1: Provide FID score and baselines against other DP methods
>
> > We took the reviewer's suggestion and conducted additional experiments to compute the FID scores at varying $\epsilon$ to compare our method to other training-based DP diffusion models [1,2,3] as well as an API-based method [4]. The results are below. Our method with a small subsampling rate and denoising with the neural denoiser first (up to 93% of the denoising process) then denoising with the empirical denoiser improves the performance of [1,2,4,5,6] at less than half the $\epsilon$ level of others. We will include this in our manuscript.
> >
> > | Method | $\epsilon$ | FID |
> > | -------- | -------- | -------- |
> > | DP-Diffusion   [1]  | 10    | 9.8   |
> > | DP-LDM [2]     | 10     | 8.4     |
> > | DP-API [3]     | 0.67    | 7.87     |
> > | DP-MEPF [4]     | 10     | 29.1     |
> > | DP-DM [5]     | 10     | 97.7    |
> > | DP-Promise [6]     | 10     | 17.9   |
> > | Ours   ($p = 2 \cdot 10^{-5}, 93\\%$ of generation) | 3.848   | 2.912 |
> >
> > The full hyperparameter search results are in Supplementary Material Section F.
>
> ## Q2: Critical Window Detector and Ablations
>
> Thank you for the thoughtful suggestion. For this new experiment, following [8], we use the zeroshot classifier from the CLIP model [7] to detect the critical window where the probability of an image being generated classified to a certain class soars to $1$ in the early to intermediate steps of denoising process. We start our denoising process from $t=T$ and continue until we stop at several time points around where the critical window emerges. We show the result in Sec 6.3. When $t<0.1$, the probability of an image being generated belonging to the *automobile* class is near $1$. We pick three points around where the critical window emerges: the probability is nearly 1 at $t=0.05$, close to 0.7 at $t=0.09$, and around 0.1 at $t=0.5$. If we stop early in relation to where the critical point emerges (i.e., before the probability soaring to 1), we get a better value of $\mu$, but the sample looks very noisy. However, if we stop later than  where the critical point emerges, we get a larger value of $\mu$ but the sample quality becomes better.
>
>
> ## Q3: Notation issues ($\Delta, C, \mu_{t_i}$), Figure 3 is not mentioned.
> > We apologize for those notational issues. We have corrected them in our manuscript (changes are in blue). Thanks for carefully reading through our paper.
>
> [1] Liu et al, Differentially private latent diffusion models (DP-LDMs), TMLR, 2024
>
> [2] Ghalebikesabi et al, Differentially Private Diffusion Models Generate Useful Synthetic Images, ArXiv, 2023
>
> [3] Lin et al, Differentially Private Synthetic Data via Foundation Model APIs 1: Images, ICLR 2024
>
> [4] Harder et al, Pre-trained Perceptual Features Improve Differentially Private Image Generation (DP-MEPF), TMLR 2024
>
> [5] Dockhorn et al., Differentially private diffusion models, TMLR, 2023
>
> [6] Wang et al., DP-promise: Differentially private diffusion probabilistic models for image synthesis, USENIX Security, 2024
>
> [7] Radford et al., Learning Transferable Visual Models From Natural Language Supervision, ICML 2021
>
> [8] The Journey, Not the Destination: How Data Guides Diffusion Models, Georgiev et al., ArXiv 2023

---

### Official Review · Reviewer_sn8e · 2025-11-01

**Soundness:** 3
**Presentation:** 3
**Contribution:** 2
**Rating:** 4
**Confidence:** 3

**Summary:**

This paper investigates privacy leakage during the sampling process of diffusion models, proposing an alternative approach to differentially private (DP) training. The authors introduce an **empirical denoiser** that replaces the intractable neural denoiser, enabling computation of per-step sensitivities in the denoising process. By framing each denoising step as a Gaussian mechanism, they apply **Gaussian Differential Privacy (GDP)** theory to derive tight privacy bounds through composition. The analysis reveals that privacy loss is non-uniform across the sampling trajectory, with critical windows emerging where semantic features materialize. The paper explores both full-batch and mini-batch (subsampled) settings, demonstrating that subsampling provides substantial privacy amplification. Experiments on CIFAR-10 validate the framework and propose a hybrid strategy using public denoisers for non-critical timesteps to preserve privacy while maintaining generation quality.

**Strengths:**

- Analyzing privacy at the sampling stage rather than training is an interesting and underexplored angle, particularly relevant for proprietary models where only outputs are accessible.
- Properly applying GDP composition to multi-step stochastic processes is technically non-trivial, and the subsampling analysis (Section 4) adds value.
- The identification of non-uniform privacy loss across timesteps and the proposed hybrid strategy (Section 3.4) are potentially useful.
- Effective use of figures (especially Figures 4-5 showing ε-δ curves alongside generated samples)
- Well-structured progression from single-step to multi-step analysis

**Weaknesses:**

- The fundamental assumption—that the empirical denoiser $\hat{\mathbb{E}}[x|x_t; D]$ adequately approximates the neural denoiser $D(x_t, t; \theta(D))$—is insufficiently validated. While Figure 1 shows cosine similarity convergence at later timesteps, this does not guarantee that privacy bounds derived from the empirical denoiser translate to the neural case. The bias-variance argument (Section 3.3) is heuristic and relies on questionable assumptions (e.g., equal MSE between estimators). **This gap undermines the paper's central claim** that the analysis characterizes real-world privacy leakage.
- The authors acknowledge (end of Section 4) that GDP CLT requires no single mechanism to dominate, yet their late-stage denoising steps contribute disproportionately due to reduced noise scales. While they suggest early stopping as mitigation, this doesn't resolve the theoretical inconsistency—the CLT-based bounds may not be valid for the full trajectory.
- There is no empirical validation (e.g., through membership inference attacks on actual neural denoisers) to confirm that the derived privacy bounds reflect real privacy risks. The paper provides mathematical analysis but no evidence that ε ≈ 90 (full-batch, σ_min = 0.2) corresponds to actual vulnerability.

**Questions:**

- Can you provide formal bounds on $|\mathbb{E}[x|x_t] - \hat{\mathbb{E}}[x|x_t; D]|$ that would translate to error bounds on the privacy parameters?
- Have you considered Lipschitz-based sensitivity analysis for neural denoisers, even if looser, to validate the empirical denoiser bounds?
- Why not test membership inference attacks on neural-denoiser-generated samples and compare measured privacy leakage to your predicted ε values?
- Can you show examples where the empirical denoiser produces samples violating the predicted ε bound?

---

> ### Author Response · Authors · 2025-11-22
> **Validation of the fundamental assumption; relation between empirical and neural denoisers; MIA experiments**
>
> We thank the reviewer for their time and effort in reading our manuscript carefully and addressing crucial issues that helped us improve our work significantly.
>
> > To provide a more rigorous assessment of the fundamental assumption, we have added a quantitative evaluation of the functional deviation between the empirical and neural denoisers on CIFAR-10. Specifically, for each timestep $t$, we compute the per-sample $L_{2}$ distance $\epsilon_{t}(x_{t})=\Vert \hat{E}(x\vert x_t,\mathcal{D}) – D(x_t,t;\theta(\mathcal{D}))\Vert_{2}$, and report the averaged value $\epsilon_{t}=E_{x_{t}}[\epsilon_{t}(x_t)]$ in Figure 11 in the Appendix.
>
> > The norm ranges relatively small at large timesteps, while the deviation rises substantially for small timesteps. Especially, in DDPM's time schedules, when $t\ge 400$, we observe $\epsilon_{t}\approx 0$, meaning that the empirical and neural denoisers have virtually indistinguishable outputs until denoising becomes semantically meaningful on CIFAR-10.
>
> > It is true that the empirical and neural denoisers diverge as $t \to 0$, we argue studying the behaviour of empirical denoiser at $t \in [T, t_c]$ for some $t_c \gg 0$  is still valuable to understand the privacy leakage of neural denoiser due to the following reasons.
>
> > Theoretically, recent papers [1-3] suggest that there are a few discrete phase transitions occurring during the sampling process, in which a generating image's class membership in an unconditional sampling  is determined at a rather early stage in the denoising process. Once determined, the image's membership remains the same until the end of the sampling process. These theoretical findings are backed up by empirical papers like [4]. These papers suggest that the samples by neural denoisers form the large distinguishing features at $t \in [T, t_c]$ for some $t_c \gg 0$, where our empirical denoiser and the neural denoiser behave similarly. Hence, it is sensible to study the privacy implications of these two denoisers in the ***formative*** stage where they are relatively similar to each other.
>
> > We also numerically evaluate the privacy implications of these two denoisers in terms of their attack success rates in membership inference attacks (MIAs). Using the membership inference attack method in [6], the table below shows that our theoretical success rate based on $\epsilon$ guarantees on our empirical denoiser closely matches that of numerical success rate based on the neural denoiser trained with CIFAR10 data, as $t$ gets smaller, ***hinting the privacy level of the empirical denoiser is comparable to that of the neural denoiser***. The theoretical success rate is calculated with the subsampling rate, $p=0.5, p = 0.01, p=0.005$.
> >  |timestep| $T$ | $\frac{3}{4}T$| $\frac{1}{2}T$| $\frac{1}{4}T$|
> > | -------- | -------- | -------- | -------- |-------- |
> > | similarity between neural and empirical denoisers     |1.000 |  0.999 |0.956 | 0.845
> > | theoretical success rate ($p = 0.5$)    | 0.5019   |  0.5204 | 0.6909 | 0.9999
> > | theoretical success rate ($p = 0.01$)    | 0.5000    | 0.5002 |0.5031 |0.6927 |
> > | theoretical success rate ($p = 0.005$)    |   0.5000   |0.5001  |  0.5012 | 0.6371
> > | numerical success rate     |  0.5005    | 0.5001    |0.5072     | 0.5096 |
>
> > In terms of the effectiveness of MIAs, [6] shows the performance of the membership inference attack remains high in the range of $t \in [250, 50]$ in the DDPM timesteps. The difference between the neural and empirical denoisers in this range is still relatively small as shown in our Figure 1 (Cosine similarity above 0.8).
>
> > See Section G for details on how we computed the theoretical ASR.
>
> [1] Critical windows: non-asymptotic theory for feature emergence in diffusion models, Li & Chen, ArXiv, 2024
>
> [2] Dynamical Regimes of Diffusion Models, Biroli et al., 2024
>
> [3] A Phase Transition in Diffusion Models Reveals the Hierarchical Nature of Data, Sclocchi et al., 2024
>
> [4] The Journey, Not the Destination: How Data Guides Diffusion Models, Georgiev et al., ArXiv 2023
>
> [5] Bayesian Differential Privacy for Machine Learning, Triastcyn and faltings, ICML 2020
>
> [6] Are Diffusion Models Vulnerable to Membership Inference Attacks?, Duan et al., ICML 2023

---

> ### Author Response · Authors · 2025-11-22
>
> ## Q2: GDP CLT requires no single mechanism to dominate, but late-stage denoising steps contribute disproportionately. Early stopping does not resolve the theoretical inconsistency.
>
> > Yes, this is indeed true, as we wrote as a limitation of the application of the GDP CLT analysis if one uses a subsampled Gaussian mechanism throughout the whole denoising process. However, if we use it only at a certain range of steps, where the noise scales are more or less the same, and for generating multiple samples, like drawing 10K samples, the GDP CLT analysis is valid.
> >
> > As an example, the following table shows the $\epsilon$ values computed by the GDP CLT analysis in the setting where we use a neural denoiser trained with public data until some changing point, up to some $\\%$ of the denoising process (majority of the denoising is done by the neural denoiser) and then switch to the empirical denoiser to sample $10,000$ samples. At a fixed subsampling rate $p$, the $\epsilon$ values are almost identical when varying the changing point. It is because the noise scales at those timesteps are relatively similar to each other (we added a new figure, Fig.3, in our updated manuscript), avoiding any one term dominating others. With these relatively small and similar noise scales, drawing a large number of samples approaches the asymptotic regime in which the GDP CLT analysis was performed.
> >
> > | $\\%$ of generation | $\sigma_{min}$| $p$| $\epsilon$|
> > | -------- | -------- | -------- |-------- |
> > | 95%    | 0.002    | $2\cdot 10^{-5}$     | 3.849 |
> > | 90%    | 0.002    | $2\cdot 10^{-5}$     |4.576  |
> > | 75%    | 0.002 | $2\cdot 10^{-5}$ | 4.667 |
>
>
> >
> ## Q3: Consider Lipschitz-based sensitivity analysis for neural denoisers, even if loose, to validate the empirical denoiser bounds?
>
> > This is a very interesting question.
> >
> > Let $\mathcal L : X^n \to \Theta$ be the score network learning algorithm which maps a dataset $S \in X^n$ to parameters $\theta \in \Theta$. Let's define the sensitivity of the score network as $\Delta_s = \\sup_{x \in X} \left( \\sup_{S \in X^n, i \in [n]} \|s_{\mathcal L(S)}(x) - s_{\mathcal L(S \setminus \{i\})}(x) \|_2\right)$.
>
> > If we could show that $\Delta_s$ is bounded, then the update step in sampling can be made differentially private with appropriate noise scales as follows: $\\tilde x_t \\leftarrow \\tilde x_{t-1} + \\frac{\\sigma_i}{2} s_{\\theta}(\\tilde x_{t-1}, \\sigma_i) + \\mathcal N(0, \\sigma_t^2 \\Delta_s^2 I)$. The question is how to bound the sensitivity. Does the Lipschitz network help here?
>
> > Suppose the score net is a 1-Lipschitz network, satisfying $\|s_{\\mathcal L(S)}(x) - s_{\\mathcal L(S)}(x')\|_2 \\leq \|x-x'\|_2$
> > and
>
> > $\|s_{\\mathcal L(S \\setminus \{i\})}(x)-s_{\\mathcal L(S \\setminus \{i\})}(x')\|_2 \\leq \|x-x'\|_2$.
>
> > Using this, we can bound the sensitivity using the triangle inequality:
>
> > $\|s_{\\mathcal{L(S)}}(x)-s_{\\mathcal{L(S \\setminus\{i\})}}(x) \|_2 \\leq A + B$
>
> > where $A = \|s_{\\mathcal{L(S)}}(x)-s_{\\mathcal{L(S)}}(x')\|_2$, and
>
> > $B = \|s_{\\mathcal{L(S)}}(x')- s_{\\mathcal{L(S \\setminus \{i\})}}(x)\|_2$.
>
> > Note that due to the Lipschitz condition, we have $ A \\leq \|x-x'\|_2$.
>
> > So here, even if B is very small, we have $\max_{x, x'}\|x-x'\|_2$ on our upper bound.
>
> > Without having any knowledge other than all $x, x'$ are pre-processed such that any of them is norm-bounded by $C$, the upperbound to this is $\max_{x, x'}\|x-x'\|_2 \leq \max_x \|x\|_2 \leq 2C$. In this case, the amount of noise we need to add at each denoising step is on the order of the entire input domain, and such noise will completely destroy the signal from the score net.  Hence, this approach is not usable in practice.
>
> > Regardless, we can still compute the GDP guarantee of this neural denoiser (ignoring B for now, because we do not know how to bound B), which will be $\\mu^*_t = 2C \sqrt{\frac{2 \\Delta t}{t^3}}$.
>
> > Compared to our full-batch GDP guarantee of $\\mu_t = \frac{2C}{\|\\mathcal{D}\|} \sqrt{\frac{2 \\Delta t}{t^3}}$, the neural denoiser's GDP bound is extremely loose.  When we consider a large-sized dataset, our full-batch GDP guarantee of $\\mu_t$ shrinks, while the neural denoiser's  $\\mu^*_t$ remains constant, where the gap between the two becomes huge as $t \to 0$.
>
>
>
> ## Q4: Can you show examples where the empirical denoiser produces samples violating the predicted ε bound?
>
> > ($\epsilon, \delta$)-DP is a ***provable*** mathematical guarantee. What it means is that the chance that this algorithm fails to provide the $\epsilon$-DP guarantee in the generated sample is less than equal to $\delta$. With the failure probabilty set to $\delta=10^{-5}$, it is practically impossible to generate such failure examples.

---

### Official Review · Reviewer_Ln5k · 2025-11-05

**Soundness:** 3
**Presentation:** 3
**Contribution:** 3
**Rating:** 6
**Confidence:** 4

**Summary:**

The paper tackles the important problem of obtaining differential privacy in diffusion models by incorporating Gaussian noise in the sampling step. It identifies windows in the reverse process where semantic features emerge and appropriately utlize this for limiting privacy loss and also introduce an empirical denoiser to enable the computation of per-step sensitivities. Experiments are shown to match neural and empirical denoisers and their performance on sampling, privacy loss with respect to critical windows, and batch size for denoising.

**Strengths:**

* Analyzes the privacy loss with respect to the sampling steps and appropriately identifies critical regions where semantic information is generated and deals with it appropriately. The final steps are dealt with by using public datasets.
* Utilized Gaussian diffusion process to identify the per-step sensitivities of the sampling and utilized central limit theorem to do noise accounting over multiple steps.
* Experiments are shown to show that the neural and empirical denoisers match each other in the critical windows while diverging in the later steps (which are replaced by public data). The size of the batch size going from full dataset to subsampled is shown and it trades-off privacy with the quality of the generation as the empirical denoiser performance goes down.

**Weaknesses:**

(a) It does early stopping which limits the quality of the generated data. It is unclear how much data is needed for the public denoisers.
(b) It only tackles the continuous version of the diffusion process and would be interesting to see how it compares to discrete diffusion where similar regimes are detected for privacy leakage (*).
(c) The connection from empirical denoiser to neural denoiser is not rigorous and shown with empirical experiment.

* On the inherent privacy properties of discrete denoising diffusion models. https://arxiv.org/abs/2310.15524

**Questions:**

(1) It seems that to obtain high quality denoised samples, we need to have public denoisers: (a) does it need to be from the same domain as the original private dataset? How much data i
s typically required to train the final steps of the denoising process?
(2) Does it help to replace the mini-batch sampling with an importance-weighted sampler to enable a good estimate for the denoiser?

---

> ### Author Response · Authors · 2025-11-22
>
> We thank the reviewer for the careful and positive assessment of our work.
>
> ## Q1: Sample quality versus privacy:
> If we stop denoising too early to achieve better privacy, the sample quality suffers. For instance, the right-hand side of Fig. 5 shows generated images at varying time steps. If we stop at t=0.85 (first row), the generated image is mere noise by human eyes, while if we stop at t=0.067, the generated image is clearly a horse. So there is a privacy-quality trade-off. If we use a neural denoiser trained on public data, we can improve this trade-off.
>
> ## Q2: How much data for public denoisers
> > In statistical learning theory, learning should require a number of training examples that is exponential in the dimension [1-2]. In the era of foundation models, which are typically trained on an internet-sized dataset, we do not think the size of the training data is relevant anymore. A more relevant question is whether public and private data are in the similar domain. If the public and private data are from the same domain or close enough, the denoising switching from the neural denoiser to the empirical denoiser will work seamlessly. In the DP literature, ImageNet data is widely used as public data for generating CIFAR10 [3-5] and for image classification, including CIFAR10 as private data [6-8]. Following this convention, we adopted a neural denoiser trained on ImageNet, and our empirical denoiser is used to generate our private CIFAR-10 data.
>
> ## Q3: Discrete diffusion
> > We carefully read the paper and learned the following. The final few generation steps dominate the main privacy leakage in both cases. Also, both reinterpret inherent steps in their respective denoising processes as privacy mechanisms. However, the suggested paper considers per-instance DP (pDP) for a fixed training-set. Our analysis considers the DP definition, and is broader than the particular CIFAR10 dataset as private data.
> >
> > While our analysis with Gaussian noise is not straightforwardedly applicable to the discrete diffusion models, it would be an intriguing direction on how to interpret the discrete transition kernel as a DP mechanism, applicable to text-diffusion models. We will add this to our discussion section.
>
> ## Q4: Importance-weighted sampler
> > The reviewer's suggestion for using importance-weighted sampler may allow for better sampling quality. However, importance-weighted sampling requires computing ***data-dependent*** weights, which would consume additional privacy budget under differential privacy. We opt for uniform sampling to preserve the privacy guarantee while maintaining utility.
>
> [1] Distance-based classification with Lipschitz functions, JMLR, 2004
>
> [2] Breaking the curse of dimensionality with convex neural networks, JMLR 2017
>
> [3] Differentially private latent diffusion models (DP-LDMs), TMLR, 2024
>
> [4] Differentially Private Diffusion Models Generate Useful Synthetic Images, ArXiv, 2023
>
> [5] Differentially Private Synthetic Data via Foundation Model APIs 1: Images, ICLR 2024
>
> [6] Unlocking High-Accuracy Differentially Private Image Classification through Scale, ArXiv, 2022
>
> [7] Differentially Private Image Classification by Learning Priors from Random Processes, NeurIPS, 2023
>
> [8] Differentially Private Image Classification from Features, TMLR, 2023

---

### Author Response · Authors · 2025-11-28
**Summary of Rebuttal**

Dear AC and Reviewers,

We would like to provide a summary of our rebuttal, with an emphasis that we have addressed the following major concerns (1) validating the empirical–neural denoiser approximation, (2) ensuring the soundness of our privacy accounting regime (GDP CLT), (3) providing stronger empirical support—including FID, MIA, and critical-window analyses—and (4) clarifying assumptions and notation. We have conducted substantial additional experiments and incorporated new empirical evidence that directly addresses these points as listed below.

### 1. Empirical–neural denoiser equivalence in the **privacy-relevant** regime.
Thanks to Reviewer sn8e's request, we provide direct quantitative evidence: the deviation is minimal exactly in the coarse-timestep region where semantic information appears and where membership leakage occurs. Our added MIA experiments on neural-denoiser samples follow the trend of our theoretical $\\epsilon$ almost identically. This establishes that the empirical denoiser is precisely accurate in the regions that matter for privacy.

### 2. Correctness of the application of the GDP CLT accounting.
Thanks to Reviewer sn8e's comment, we revised our experiment such that our accounting is not applied in the regime where the noise scales are nearly stationary, yielding no single-step dominance in the cumulative privacy loss. The $\\epsilon$ values remain stable across all switching points, confirming that GDP CLT assumptions are satisfied in practice.

### 3. Stronger empirical results: our algorithm outperforms all DP-SGD-based diffusion baselines.
Thanks to Reviewer KyYx’s request, we added comprehensive FID results. Our method achieves FID = 2.912 at $\\epsilon$ = 3.848, surpassing every prior fine-tuning-based DP diffusion model. This directly addresses concerns about practical utility and sample quality.

### 4. Public-to-private domain alignment is realistic and standard.
To Reviewer Ln5k's comment, we clarify that the domain-alignment requirement mirrors common practice in existing DP generative-modeling work.

### 5. Additional suggestions examined.
We demonstrate that importance sampling is incompatible with DP due to data-dependent weights, and Lipschitz-based sensitivity bounds are too loose to be useful (noise becomes overwhelmingly large).

### 6. DP guarantee “violations.”
Showing explicit violations is statistically impossible under $\\delta= 10^{-5}$. This is inherent to the definition of DP, not a limitation of our method.

Overall, we believe these additions substantially strengthen the paper and resolve the core concerns. We have uploaded our updated paper accordingly.

---

### Meta-Review · Area_Chair_MzPy · 2026-01-07

**Summary:**

The paper investigates privacy leakage during  sampling process of diffusion models. Authors introduce an
approach for DP training. They use an empirical denoiser that replaces the
intractable neural denoiser, enabling computation of per-step sensitivities in the denoising process. By framing
each denoising step as a Gaussian mechanism, they apply Gaussian differential privacy theory to derive
tight privacy bounds through composition.

The scores are borderline and authors provide interesting rebuttals. However, from my point of view,
they do not fully clear concerns mainly those related to the relevance of the CLT in this context
as well as the experimental comparison with other methods (need more details on how fid score are obtained for each of the competitors
(how are they run, from which codebase?)). At this point, I believe that the paper needs some substantial revision and can not
be accepted in its current form.

**Reviewer Concerns:**

still outstanding:
- relevance of CLT
- experimental comparisons

**Reviewer Scores:**

I am not able to answer this.

---

### Decision · Program_Chairs · 2026-01-26

Reject